# The Homocysteine and Metabolic Syndrome: A Mendelian Randomization Study

**DOI:** 10.3390/nu13072440

**Published:** 2021-07-16

**Authors:** Ho-Sun Lee, Sanghwan In, Taesung Park

**Affiliations:** 1Interdisciplinary Program in Bioinformatics, Seoul National University, Seoul 08826, Korea; hs4369@gmail.com; 2Forensic Toxicology Division, Daegu Institute, National Forensic Service, Andong-si 39872, Gyeongsangbuk-do, Korea; swin@korea.kr; 3Department of Statistics, Seoul National University, Seoul 08826, Korea

**Keywords:** homocysteine, mendelian randomization, metabolic syndrome

## Abstract

Homocysteine (Hcy) is well known to be increased in the metabolic syndrome (MetS) incidence. However, it remains unclear whether the relationship is causal or not. Recently, Mendelian Randomization (MR) has been popularly used to assess the causal influence. In this study, we adopted MR to investigate the causal influence of Hcy on MetS in adults using three independent cohorts. We considered one-sample MR and two-sample MR. We analyzed one-sample MR in 5902 individuals (2090 MetS cases and 3812 controls) from the KARE and two-sample MR from the HEXA (676 cases and 3017 controls) and CAVAS (1052 cases and 764 controls) datasets to evaluate whether genetically increased Hcy level influences the risk of MetS. In observation studies, the odds of MetS increased with higher Hcy concentrations (odds ratio (OR) 1.17, 95%CI 1.12–1.22, *p* < 0.01). One-sample MR was performed using two-stage least-squares regression, with an *MTHFR* C677T and weighted Hcy generic risk score as an instrument. Two-sample MR was performed with five genetic variants (rs12567136, rs1801133, rs2336377, rs1624230, and rs1836883) by GWAS data as the instrumental variables. For sensitivity analysis, weighted median and MR–Egger regression were used. Using one-sample MR, we found an increased risk of MetS (OR 2.07 per 1-SD Hcy increase). Two-sample MR supported that increased Hcy was significantly associated with increased MetS risk by using the inverse variance weighted (IVW) method (beta 0.723, SE 0.119, and *p* < 0.001), the weighted median regression method (beta 0.734, SE 0.097, and *p* < 0.001), and the MR–Egger method (beta 2.073, SE 0.843, and *p =* 0.014) in meta-analysis. The MR–Egger slope showed no evidence of pleiotropic effects (intercept −0.097, *p =* 0.107). In conclusion, this study represented the MR approach and elucidates the significant relationship between Hcy and the risk of MetS in the Korean population.

## 1. Introduction

During recent decades, metabolic disease has become a major health concern worldwide with the spread of the Western diet and lifestyle, and the increase in the elderly population. Metabolic syndrome (MetS) is defined by WHO as a pathologic condition characterized by hypertension, glucose abnormalities, central obesity, and hyperlipidemia [1]. The global prevalence of overweight and obesity has continuously been growing and has now reached epidemic proportions [2]. With this phenomenon, cardio-metabolic abnormalities and MetS are expected to become more prevalent in youth, as well.

Nutrient intake is known as an important lifestyle factor for non-communicable disorders [3]. Homocysteine (Hcy) is an amino acid intermediate formed during the metabolism of the essential amino acid methionine. Hcy can be recycled into methionine with the aid of vitamin B_12_ and folic acid, or converted into cysteine with vitamin B_6_ as a cofactor. Hyperhomocysteinemia exerts a wide range of biological effects on multiple organs and is known to be associated with a number of aging-related diseases, including cardiovascular disease, dementia, neural tube defects, and cancer, through different mechanisms such as vascular dysfunction [4,5,6,7]. In addition, it has been suggested that an elevated Hcy level is patho-physiologically involved in the increased risk of MetS [8]. However, the mechanisms involved in Hcy-associated diseases have not been fully elucidated.

Mendelian randomization (MR), an established useful tool, provides an opportunity for elucidating the causal effect of an exposure on an outcome using genetics within the framework of an observational setting [9]. Three key assumptions of an instrumental variable (IV) behind MR studies must be considered for it to be applied appropriately: (1) the genetic variants must influence the exposure of interest; (2) the genetic variants must not affect the outcome directly, but only potentially indirectly via the exposure; (3) the genetic variants that influence the exposure must not associate with any potential confounding factors. Therefore, MR utilizes IVs such as genetic variants that act as proxies for environmental factors to assess the causal relationship between an exposure and an outcome of interest. These genetic variants are randomly assigned during meiosis, yielding a random distribution of genotypes in study populations. Genetic variants may cause the outcome or exposure. Thus, MR is often robust to the issue of confounding and reverse causation inherent in observational epidemiologic studies. Recent genome-wide association studies (GWAS) identified single nucleotide polymorphisms (SNPs) influencing MetS in the Korean population [10], which may be able to investigate a potential causal role of Hcy in MetS using the MR approach.

The aim of our study is to assess the causal influence of Hcy on MetS in adults with the use of MR. We analyzed data of the Korean Genome and Epidemiology study (KoGES) Consortium, which includes multiple independent prospective cohorts differing based on residential areas of the participants: the Health Examinees (HEXA) study, the Cardiovascular Disease Association Study (CAVAS), the Korea Association Resource (KARE) study. We analyzed one-sample MR in 5902 individuals (2090 MetS cases and 3812 controls) from the KARE and two-sample MR from the HEXA (676 cases and 3017 controls) and CAVAS (1052 cases and 764 controls) datasets to evaluate whether genetically increased Hcy level influences the risk of MetS. One-sample MR was performed using two-stage least-squares regression and weighted Hcy generic risk score as an instrument. Two-sample MR was performed with five genetic variants selected by the GWAS as the instrumental variables. Using one-sample MR, we found an increased risk of MetS. Two-sample MR supported that increased Hcy was significantly associated with increased MetS risk by using the inverse variance weighted method the weighted median regression method and the MR–Egger method in meta-analysis.

## 2. Materials and Methods

### 2.1. Study Population of Exposure and Outcome Data

We used one-sample MR and two-sample MR approaches by using GWAS data with participants from the KoGES Consortium. Exposure data were obtained from the KARE cohort, which was the fifth 2-year follow-up phase, in 2011–2012 (Ansan-Ansung community-based cohort study). Its study design, sampling, concept, and consent are described in a previous study [11]. Among the whole cohort population (*n* = 8840), Hcy data were available in 6267 individuals. After excluding missing Hcy data, we tested the causal effect of blood Hcy in 5902 individuals from 2090 cases and 3812 controls for exposure data. Outcome data of two independent prospective cohorts on MetS were retrieved from the HEXA (676 cases and 3017 controls) and CAVAS (1052 cases and 764 controls). Detailed information on the studies is summarized in Appendix A. We obtained anonymous health records and information on social history, lifestyle, diet, and daily activities provided by the National Biobank of Korea, the Korea Disease Control and Prevention Agency, Republic of Korea. The present study was approved by the institutional review board of Seoul National University (E1908/001-004). 

### 2.2. Diagnosis of Metabolic Diseases

MetS is defined by the presence of three or more of the following five components, according to the NCEP-ATP III criteria, except for the determination of central obesity [10]. Waist circumference cut-off value was based on the report by the Korean Society for the Study of Obesity that central obesity is given as waist-high circumference (≥90 cm for men and ≥85 cm for women). The details of other MetS criteria have been described in [12]. MetS score was calculated for each subject, as the summation of the number above the cut-off, for each MetS component, ranging from 0 to 5. The hypertension is defined as systolic/diastolic pressure ≥130/85 mmHg or antihypertensive drug treatment.

### 2.3. Instrumental Variables

The genotypes were derived from the Affymetric Genome-Wide Human SNP Array 5.0 Chip, which contains approximately 420,000 variants. Details on the quality control process have been published previously [10]. We filtered out variants whose missing rates were larger than 0.01, being monomorphic, and whose *p*-values of Hardy Weinberg Equilibrium test results were below *p*  <  1 × 10^−6^ [10]. The individual variant was recoded as 0, 1, or 2 according to the number of trait-increasing alleles. The selection of the SNPs modifying Hcy levels to be used as instruments in our study was based on loci achieved with a Bonferroni-corrected significance of *p* <  4 × 10^−8^. The corresponding effect estimate and standard errors of the SNPs were obtained. Genotyping and quality control procedures are described in Appendix A.

### 2.4. Statistical Analyses

For the characteristics of participants, data were presented as mean (standard deviation) or median (interquartile range) depending on the distributed normality, or as percentages (%) for categorical variables for the characterization of subjects (Table 1). The participants were classified into quartiles according to their log-transformed blood Hcy level, then we analyzed distinction by one-way analysis of variance or Kruskal–Wallis test for continuous variables or chi-square test for categorical variables. 

To identify independently associated loci, we used LD-clumping with an *r*^2^ threshold of 0.01 to select a set of independent instruments for Hcy by TwoSampleMR package. We assessed F-statistics for checking weak instrument bias. All analyses were adjusted for age, sex, and regional area. A logistic regression model was applied to calculate the odds ratio (OR) of MetS for individual SNP and selected SNPs as IVs. Heterogeneity was measured between variant-specific causal estimates by Cochran Q-derived IVW estimate, and the MR–Egger slope was detected for the directional pleiotropic effects. To test for association between confounding factors and each SNP, we performed linear or logistic regression of confounders against genotype (coded as 0, 1, or 2; additive genetic model).

With one-sample MR analyses, the causal effect of the Hcy on MetS can be estimated by using 2-stage least-squares (2SLS) regression [13]. In the first stage, Hcy was regressed on the genetic instrument, which is the *MTHFR* C677T variant from the imputed dataset or other SNPs with genetic risk scores (GRS) based on 5 selective SNPs. We constructed a weighted Hcy-increasing GRS by summing the number of Hcy-increasing alleles under an additive model weighted by the effect sizes of the variants estimated. In the second stage, the MetS is regressed over the predicted values of the Hcy by using logistic regression. The β-coefficient from the second stage can be interpreted as the change in the MetS risk per SD increase in the Hcy level due to the IV.

For the two-sample MR analysis, the estimation of the causal effect of risk factors on MetS was analyzed by the inverse variance weighted (IVW) analysis and the weighted median regression and MR–Egger methods. *p*-values < 0.05 were considered statistically significant. All statistical analyses were performed using R Software (Version 2.14.0; R Foundation for Statistical Computing, Vienna, Austria).

## 3. Results

### 3.1. Characteristics of Study Participants

The characteristics of the observation study participants are presented in Table 1. The participants consisted of 5902 individuals (log Hcy < 2.39 umol/L, *n* = 1479; 2.39 ≤ log Hcy < 2.56 umol/L, *n* = 1475; 2.56 ≤ log Hcy < 2.74 umol/L, *n* = 1470; log Hcy > 2.74 umol/L, *n* = 1478). Participants with higher Hcy levels were of older age and there was a higher frequency of smoking with men than women. There were no significant differences in the BMI between quantiles of Hcy. The physical activity (PA) was obtained from the metabolic equivalent of task (MET) score. The METs (hrs/week) were calculated by summing each type of activity (1.0 for sedentary, 1.5 for very light, 2.4 for light, 5.0 for moderate, and 7.5 for intense) [14]. Dietary habits were assessed using a recommended food score (RFS), which is based on reported consumption of foods bearing high amounts of antioxidant nutrients, consistent with the current American dietary guidelines [15]. We used the modified RFS that follows the current Korean food guidelines adapted to the Korean diet [16]. We identified that the blood homocysteine levels were associated with a decrease in the RFS score. There were no significant differences in the PA between the Hcy quartiles. In addition, those in the upper quartiles of Hcy were likely to have a higher MetS count and more history of type 2 diabetes than those in the lower quartiles of Hcy.

### 3.2. Observational Analysis for Association between MetS and Hcy and Other Variables

Univariate logistic regression analysis demonstrated that higher levels of Hcy were associated with MetS with the odds ratio (OR) 1.06 (95% confidence interval (CI) 1.05–1.08, *p* < 2 × 10^−16^) (Table 2). In multivariate analyses, higher levels of Hcy were an independent predictor of MetS after adjustment for (1) age, sex, and area (OR 1.03, 95% CI 1.02–1.05, *p* = 1.53 × 10^−7^); (2) age, sex, area, smoking, and drinking (OR 1.03, 95% CI 1.02–1.04, *p* = 2.97 × 10^−7^); and (3) age, sex, area, smoking, drinking, RFS, and BMI (OR 1.13, 95% CI 1.09–1.18, *p* = 7.03 × 10^−9^). The odds ratio remained unchanged. Therefore, we selected age, sex, and area as covariates. We also assessed the association between MetS and other environmental factors. We found that there were no associations between MetS and other potential confounders (smoking, drinking, RFS, and PA) except for BMI (Appendix A). PA was not associated with both Hcy and MetS.

### 3.3. Instrumental Variable Selection

Twenty-seven SNPs associated with Hcy concentration in the GWAS were used as instrumental variables (Appendix A) based on a Bonferroni-corrected significance, regardless of evidence of a functional impact of the SNP on Hcy concentration. We then added SNPs rs1801131 and rs1801133, which were polymorphisms in the *MTHFR* gene and known to have the strongest effect on the serum Hcy in the general population. The information of SNPs was from the imputed dataset with IMPUTE2 using the JPT/CHB component of HapMap. For further MR analysis, we selected five SNPs (rs12567136, rs1801133, rs2336377, rs1624230, and rs1836883) based on linkage disequilibrium (LD) (as assessed by r^2^ < 0.1) among the 27 associated SNPs. An F statistic was very high for all genetic variants which were strong instruments (F = 241.2 for all combined instruments). Characteristics of these SNPs and their association with phenotypes are summarized in Table 3 and Appendix A.

We investigated the association between each of the five selected SNPs and other confounding factors (smoking, alcohol consumption, dietary habits (RFS), and BMI). There were no confounding factors associated with IVs (Table 4).

We also analyzed the association between the five SNPs and each component of MetS, and the odds ratio of SNPs for each component of MetS. As a result, only two SNPs, rs2336377 and rs1801133, were found to be associated with high blood pressure and high triglyceride level, respectively (Appendix A). Thus, this component-wise analysis shows that the five SNPs have no or selective effects on MetS components, which resulted in no direct association with MetS.

### 3.4. One-Sample MR

To assess one-sample MR using the *MTHFR* C677T variant for causality of association between Hcy and MetS, we calculated an MR estimate of the effect of the plasma Hcy levels on the risk of MetS (OR_MetS/Hcy_) as log OR_MetS/Hcy_ = (log OR_MetS/per T-allele_)/ β_Hcy/per T-allele_, as in previous studies [17,18]. Log OR_MetS/Hcy_ is the (log) increase in MetS risk by SD unit increase in the natural log-transformed plasma Hcy (MR estimate). β_Hcy/per T-allele_ is the number of SD differences in the Hcy levels per allele (SD/ allele). The standard error of the MR estimate was derived using the Delta method [19]. We observed that each 1-SD increase in the natural log-transformed plasma Hcy level was significantly associated with a 2.07-fold increased risk of MetS (95% CI: 1.05–27.35, *p* = 0.044). 

The genetic risk score (GRS) comprising five SNPs was approximately normally distributed within the KARE dataset. Results from MR analysis using weighted GRS as IVs for Hcy were consistent with the observational analyses, providing evidence that increased Hcy caused a higher risk of MetS. When checking the assumptions of 1SMR, we found evidence of an association between the weighted GRS and MetS. Using weighted GRS with five SNPs, the highest OR was observed in the dominant model (CC vs CT or TT genotype; OR = 3.93, 95% CI = 3.074–5.026, *p* = 0.043).

### 3.5. Two-Sample MR

With 27 SNPs, we identified that estimated the potential causal effect of Hcy on MetS was significant in two Korean cohorts (0.735–1.024 SD change in MetS per 1 SD higher Hcy depending on methods). Using five genetic variants based on LD-clumping, we found that increased Hcy was significantly associated with increased MetS risk using weighted median regression (estimate (95% CI),0.73 (0.54–0.92); *p* < 0.01) and IVW (beta (95% CI), 0.72 (0.50–0.94); *p* < 0.01) by HEXA and CAVAS cohorts (Table 5). The MR–Egger method also showed that Hcy increased the risk of MetS (beta (95% CI), 2.07 (0.42–3.73); *p* = 0.01). It showed evidence of low heterogeneity (Cochran Q = 8.696, *p* = 0.10). There was no evidence of directional pleiotropy with five variants from the MR–Egger regression analysis (intercept = −0.097, *p* = 0.107, I^2^_GX_= 98.5%). A high value of I^2^_GX_ suggests that the instrument effect sizes are estimated well, and that measurement error/weak instrument bias is unlikely to affect the results of standard MR–Egger analyses [19].

## 4. Discussion

Previous observational studies of Hcy have yielded inconsistent results, some associating Hcy with hypertension, one of the components of MetS [20], and others failing to identify such association [21]. Inferring causal effects from classical observational studies may be problematic because of unmeasured confounding factors or reverse causality for identifying risk factor of disease. MR studies between Hcy and T2D or coronary artery disease have been well conducted [22,23,24]. However, an association between Hcy and MetS using MR approach compared to previous studies has not been identified. So far, coffee intake, C-reactive protein, vitamin D, and uric acid were assessed for the causal relationship with MetS using MR [20,25,26,27]. To our knowledge, this is the first study demonstrating that elevated Hcy may have a causal role in the development of MetS.

Many studies support the contributions of the *MTHFR* C677T polymorphism to folic acid metabolism and blood Hcy levels [28,29]. Methylenetetrahydrofolate reductase (MTHFR) is a key enzyme of Hcy that catalyzes the conversion of Hcy to methionine. The MTHFR C677T allele results in an amino acid change, and a reduction in MTHFR activity leads to hyper-homocysteinemia, which is potentially an independent risk factor for myocardial infarction, hypertension, and stroke [30,31]. Therefore, we chose the *MTHFR C677T* polymorphism as a target SNP for one-sample MR, which is the standard implementation of MR in a single data set on the SNPs, exposure, and outcome for all participants.

We found that Hcy is associated with an increased risk of MetS (OR 2.07 per 1-SD Hcy increase). In addition, our IV (*MTHFR* C677T) was associated with homocysteine concentration, with an *F* statistic = 208 (*p* = 6 × 10^−46^; crude model), indicating that weak instrument bias is unlikely to be substantially influencing our analyses. Even though the frequency of this risk allele is variable (Han Chinese 0.47, East Asian 0.29, European 0.36, African 0.09, American 0.47, and South Asian 0.12 from 1000 Genomes Project Phase 3), our data had 87% (OR = 2.07) power to detect the causal odds ratio (Type 1 error rate 0.05) according to online sample size and power calculator. Consistent with this finding, weighted GRS with five SNPs per SD increase in Hcy (μmol/L) was associated with an increase in odds of MetS (OR = 3.93; 95% CI = 3.074–5.026; *p* = 0.043). By applying the two-sample MR approach, the causal effect estimates of Hcy levels on MetS across the individual SNPs confirmed again that increased Hcy was significantly associated with increased MetS risk using weighted median regression (estimate (95% CI),0.73 (0.54–0.92); *p* < 0.01) and IVW (beta (95% CI), 0.72 (0.50–0.94); *p* < 0.01) by two Korean cohorts (Table 4). The MR–Egger method also showed that Hcy increased the risk of MetS (beta (95% CI), 2.17 (0.87–3.47); *p* < 0.01).

We investigated the association between IVs and other confounding factors (smoking, alcohol consumption, dietary habits, and BMI) for MetS. Furthermore, we found that each IV was not linked to the confounders we measured. Given these results, confounding factors, including dietary habits, were not associated with IVs which were found to be associated with MetS through Hcy in this study. However, it is difficult to rule out unmeasured or unknown confounders that can affect the association between Hcy and MetS.

The association between Hcy and MetS risk remains poorly understood. Several possible explanations have been proposed to offer some mechanistic insights. Hyperhomocysteinemia has been proposed as being part of the pathophysiology of cardiovascular disease due to its various biological effects, such as vascular damage, oxidative stress-induced DNA damage [32], neuronal apoptosis [33], cell cytotoxicity [34], and endothelial nitric oxide production [35]. Homocysteine acts as a methyl donor when it is converted to S-adenosyl-methionine, and a recent study demonstrated an association between the Hcy and DNA methylation in cardiovascular disease and dementia [36,37] with *MTHFR* C677T polymorphism, which suggests that Hcy also might play a role in the pathogenesis of those diseases via alterations in DNA methylation. However, the role of Hcy in the development of metabolic disease is unclear.

Our study had several limitations. Firstly, some epidemiological studies suggested that the pathogenesis of NAFLD and MetS seems to have common pathophysiological mechanisms, with focus on insulin resistance and obesity as key factors [38,39,40]. There was strong evidence for genetic determinants of each MetS and NAFLD [41]; however, few studies have investigated the causal effect of Hcy on both NAFLD and MetS components. Since Hcy levels were involved in the development of non-alcoholic fatty liver disease (NAFLD) [42], there is a need to investigate the underlying mechanisms linking Hcy-associated NAFLD and MetS [43]. However, there are no such data available for NAFLD. Thus, the genetic association between NAFLD and Hcy for the development of MetS is not feasible in the present study. Further studies on the Hcy-associated MetS and NAFLD using an MR approach are warranted to identify more relevant genes for understanding etiology of metabolic disease. Secondly, our finding was conducted in a Korean population, and therefore, this study might not be generalized across populations. However, this could avoid the potential bias that might be caused by differences in genetic background. In addition, it is difficult to completely exclude the influence of potential directional pleiotropy. The causal effect estimates of Hcy on MetS across the individual SNPs showed low heterogeneity (Q  =  8.696, *p * =  0.10), but no evidence of a pleiotropic effect through MR–Egger intercept test (Egger intercept for Hcy = 0.097, *p* = 0.107 for five SNPs). However, we caution the interpretation of the sensitivity analyses, due to the small number of SNPs. Besides, due to a limited number of individuals included in our study, our results should be further confirmed and strengthened by other validation studies, using larger cohorts. Lastly, one of the predominant molecular mechanisms of Hcy in the human body is reported to be related to folate and methionine cycles through transmethylation pathway [3]. We need to consider the epigenetic mechanism in modifying DNA methylation without genetic changes or interplay between genetic and epigenetic mechanisms that may therefore lead to increased risk of MetS. Nonetheless, MR studies can provide reliable evidence for the effect of modifiable risk factors on disease and can overcome some of the limitations of observational studies [44].

Recent advances in large-scale genetic studies provide thousands of genetic variants that underlie complex diseases, leading to a better understanding of the genetic architecture of the diseases. Mendelian randomization shows the potential use of observational epidemiological studies with genetic availability along with biological knowledge to investigate the causal relationship between exposure and outcome. In this study, we identified the SNPs affecting Hcy, not directly MetS, suggesting that the associated genetic variants might provide information on the biological mechanisms of MetS. Hcy might be a functional intermediate to understand the biological process through which genetics affect MetS [45,46]. The strength of the causal relationship between modifiable exposure and risk of disease identified by MR can also help improve the drug target identification or drug development. An understanding of the causal role of Hcy in MetS patients and its risk factors including obesity might be relevant because Hcy concentration can be effectively lowered by simple, safe, and inexpensive interventions, such as supplementation with folic acid and vitamin B.

## 5. Conclusions

We provide evidence by implementing a comprehensive MR study design that there is a causal link between Hcy and increased MetS risk. We expect that our results might provide the effect of Hcy exposure on MetS adjusting for potential genetic confounders. The findings from our study warrant further research to uncover the mechanism that implicates Hcy and metabolic-related traits in MetS onset.

## Figures and Tables

**Table 1 nutrients-13-02440-t001:** Characteristics of study subjects in the KARE cohort.

Variables	All	Quartile of Blood Homocysteine Concentration (umol/L) ^1^				*p*-Value
		Q1 (<2.39)	Q2 (2.39–2.56)	Q3 (2.56–2.74)	Q4 (>2.74)	
Sample size, *n*	5902	1479	1475	1470	1478	
Age, year	58 (52,67)	54 (51,62)	57 (52,65)	59 (53,71)	64 (55,71)	<0.01
Female, %	53	81.7	58.5	41.8	29.8	<0.01
Area (Urban), %	51.3	54.5	54.7	51.7	44.2	<0.01
BMI, kg/m^2^	24.44 (3.11)	24.07 (22.29,26.05)	24.31 (22.51,26.44)	24.43 (22.53,26.4)	24.37 (22.37,26.2)	0.54
Smokers ^2^, %	15.2	7.3	11.7	18.1	23.8	<0.01
Drinkers ^3^, %	43.7	33.5	43.6	45.7	52.1	<0.01
PA, METs (h/week)	92.94 (13.3)	93.08 (12.19)	93.05 (13.17)	93.06 (13.58)	92.52 (14.41)	0.47
RFS, score	19 (13,25)	21 (15,27)	20 (14,25)	19 (12,25)	18 (11,24)	<0.01
WC, cm	84.45 (9.2)	82 (75.18,87.85)	83.9 (77.80,89.52)	85.24 (9.05)	86.5 (8.95)	<0.01
FAG, mmol/L	95 (89,105)	93 (88,102)	94 (89,104)	96 (90,106)	97 (91,107)	<0.01
TG, mmol/L	120 (87,172)	111.5 (82,157)	117 (87,167)	124 (90,175)	129 (91,190.75)	<0.01
HDL-C, mmol/L	42 (36,49)	44 (37,51)	43 (37,50)	42 (36,49)	40 (34,47)	<0.01
BP, mmHg						
Systolic	118 (108,128)	119 (109,129)	117 (107,127)	119 (109,129)	120 (110,131)	<0.01
Diastolic	75 (69,81)	72 (68,79)	75 (70,81)	76 (70,82)	76 (70,82)	<0.01
MetS, %	35.4	24.7	30.9	37.8	48.4	<0.01
T2D, %	21.4	16.8	19.2	21.6	28.1	<0.01

^1^ logarithmic values; ^2^ current smokers; ^3^ current drinkers. *p*-values were calculated by ANOVA or Kruskal–Wallis test for continuous variables and chi-square test for categorical variables. Continuous variables were described as mean (standard deviation) or median (interquartile range) or depending on the distributed normality, and categorical variables were described as %. BMI, body mass index; PA, physical activity; METs, metabolic equivalent of tasks; RFS, recommended food score; FAG, fasting glucose; WC, waist circumference; TG, triglyceride; HDL-C, high-density lipoprotein cholesterol; MetS, metabolic syndrome; T2D, type 2 diabetes.

**Table 2 nutrients-13-02440-t002:** The association between metabolic syndrome and homocysteine in the KARE cohort.

	Univariate		Multivariate 1		Multivariate 2		Multivariate 3	
	OR (95% CI)	*p*-Value	OR (95% CI)	*p*-Value	OR (95% CI)	*p*-Value	OR (95% CI)	*p*-Value
Hcy	1.06 (1.05–1.08)	<0.01	1.03 (1.02–1.05)	<0.01	1.03 (1.02–1.04)	<0.01	1.03 (1.02–1.05)	<0.01

Multivariate 1: adjusted for age, sex, area; Multivariate 2: adjusted for age, sex, area, smoking, drinking; Multivariate 3: adjusted for age, sex, area, smoking, drinking, RFS, BMI.

**Table 3 nutrients-13-02440-t003:** Effect estimates for associations of genetic instruments with Hcy and MetS.

SNP(f)	Chr	Gene	EA	Association with Log-Transformed Hcy ^a^			Association with MetS ^b^		
				Beta	SE	*p*-Value	OR	95% CI	*p*-Value
rs12567136	1	*CLCN6*	A	−0.058	0.008	3.23 × 10^−10^	0.998	0.976,1.020	0.867
rs1801133	1	*MTHFR*	G	−0.039	0.005	2.74 × 10^−17^	0.949	0.850,1.059	0.354
rs2336377	1	LOC390997	G	−0.045	0.008	1.74 × 10^−8^	1.005	0.988,1.022	0.577
rs1624230	3	*KNG1*	C	−0.027	0.005	3.81 × 10^−8^	0.998	0.981,1.014	0.774
rs1836883	11	*NOX4*	T	0.029	0.005	9.55 × 10^−10^	0.983	0.967,0.998	0.031

EA, effect allele; Chr, chromosome; Hcy, homocysteine; MetS, metabolic syndrome. ^a^ Results were derived from observation study of the KARE cohort and adjusted for age, sex, and regional area. ^b^ Results were derived from merged data of the HEXA and CAVAS cohorts and were adjusted for age, sex, and study site.

**Table 4 nutrients-13-02440-t004:** The association between each instrumental variable and confounding factors.

Instrumental Variable	Confounding Factors (Beta (Standard Error), *p*-Value)			
	Smoking	Alcohol Consumption	Dietary Habits (RFS)	BMI
rs12567136	0.22(0.32),0.50	−0.11(0.29),0.70	1.21(1.03),0.24	−0.05(0.38),0.89
rs1801133	0.02(0.11),0.85	0.01(0.08),0.91	−0.54(0.28),0.05	−0.05(0.11),0.62
rs2336377	−0.06(0.12),0.60	0.03(0.09),0.72	0.10(0.31),0.75	0.11(0.12),0.35
rs1624230	−0.15(0.13),0.25	0.12(0.10),0.22	0.12(0.34),0.73	−0.19(0.13),0.15
rs1836883	0.16(0.11),0.13	0.03(0.08),0.73	−0.40(0.27),0.15	−0.08(0.10),0.44

RFS, recommended food score; BMI, body mass index; Smoking: current smoker vs non-, ex-smoker; Drinking: current drinker vs non-, ex-drinker. Analyses were adjusted for age, sex, and area. Genetic model is additive.

**Table 5 nutrients-13-02440-t005:** Estimates from MR methods for the association between Hcy and MetS.

Number of IVs	Methods	Beta Coefficient	95% CI	*p*-Value ^1^
27 SNPs	Weighted median	0.735	0.548, 0.923	<0.001
	IVW ^2^	0.741	0.588, 0.894	<0.001
	MR–Egger ^3^	1.024	0.391, 1.656	0.002
5 SNPs	Weighted median	0.734	0.544, 0.923	<0.001
	IVW ^4^	0.723	0.496, 0.941	<0.001
	MR–Egger ^5^	2.073	0.421, 3.725	0.014
	Weighted GRS ^6^	1.369	1.123, 1.615	0.043

^1^ Adjusted for age, sex, and study site. IV, instrumental variable; ^2^ Q = 22.58, *p* = 0.707; ^3^ intercept: −0.019, *p* = 0.367, I^2^_GX, 87.1%; ^4^ Q = 8.696, *p* = 0.10; ^5^ intercept: −0.097, *p* = 0.107, I^2^_GX, 98.5%; ^6^ derived from one-sample MR.

## Data Availability

GWAS dataset and epidemiological data for KoGES are third-party data and are available under the approval of the data access committee of the National Biobank of Korea (http://www.nih.go.kr/NIH/eng/contents/NihEngContentView.jsp?cid=65714&menuIds=HOME004-MNU2210-MNU2327-MNU2329-MNU2338, last accessed 25 January 2021).

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
