# Peer review of "The Homocysteine and Metabolic Syndrome: A Mendelian Randomization Study"

_nutrients, 2021, doi:10.3390/nu13072440_

Round 1

Reviewer 1 Report

The manuscript (nutrients-1293997-peer-review-v1) entitled “The Homocysteine and Metabolic Syndrome: A Mendelian Randomization Study” is a resubmission of a previous version manuscript (nutrients-1293997-peer-review-v1). This time, there authors, Ho-Sun Lee, Sanghwan In, and Taesung Park, contributed to this paper. The authors used Mendelian Randomization (MR) to investigate the causal influence of Homocysteine (Hcy) on the metabolic syndrome (MetS). They used three data bases and evaluated whether genetically increased Hcy level influences the risk of MetS.

Mets is defined by the presence of three of the five characteristics, increased blood pressure, high blood sugar, excess body fat around the waist, and abnormal cholesterol or triglyceride levels. It is very helpful if individual genetic variations can be linked to a particular characteristic of MetS. Those 5 SNDs listed in table 3 are not directly associated with MetS. To better understand the mechanism, please analyze the associations of the 5 SNPs with the individual characteristics of MetS. Please discuss why a direct association could not be established.

Reviewer 2 Report

Indeed, the last version presented by authors is ameliorated mainly concerning the development  and interpretation of the statistical results 

Author Response

This manuscript is a resubmission of an earlier submission. The following is a list of the peer review reports and author responses from that submission.

Round 1

Reviewer 1 Report

The manuscript (nutrients-1236435) entitled “The Homocysteine and Metabolic Syndrome: A Mendelian 2 Randomization Study” is a research article by Ho-Sun Lee, and Taesung Park. The authors used Mendelian Randomization (MR) to investigate the causal influence of Homocysteine (Hcy) on the metabolic syndrome (MetS). They used three data bases and evaluated whether genetically increased  Hcy level influences the risk of MetS.

The association of Hcy and MetS has been know for a while. The conclusion has  been published in a lot of papers. Using another method to demonstrate a well-known conclusion has limited value, especially for understanding the mechanisms.

Reviewer 2 Report

Authors studied a key molecule  in the onset and progression of atherosclerosis, without considering that it is strictly linked to NAFLD as evident in...Plasmatic higher levels of homocysteine in Non-alcoholic fatty liver disease (NAFLD). Nutr J 12, 37 (2013). https://doi.org/10.1186/1475-2891-12-37.

Now, being NAFLD the main morbidity associated with the Metabolic syndrome, even though the inner mechanisms underlying this disease are far from being clarified, as evident in...J. Clin. Med. 2020, 9(1), 15; https://doi.org/10.3390/jcm9010015, authors should present relevant data on association of this illness with homocysteine in their patients with  the Metabolic syndrome

In case there are no available data, authors should comment at large on this key aspect, putting this lack of data as a main limitation.

Authors are kindly requested to control the normality of distribution (by S-W test) of some variable, having some of them very large SD.

In Table 1 the statistics corresponding to the three cohorts is lacking.

Authors should fill this gap.

Round 2

Reviewer 1 Report

The authors have not addressed my comments.

Reviewer 2 Report

Authors correctly answered comments